# A Video Summarization Model Based on Deep Reinforcement Learning with Long-Term Dependency

**DOI:** 10.3390/s22197689

**Published:** 2022-10-10

**Authors:** Xu Wang, Yujie Li, Haoyu Wang, Longzhao Huang, Shuxue Ding

**Affiliations:** School of Artificial Intelligence, Guilin University of Electronic Technology, Jinji Road, Guilin 541004, China

**Keywords:** video summarization, reinforcement learning, unsupervised learning, long-term dependency, auxiliary summarization loss

## Abstract

Deep summarization models have succeeded in the video summarization field based on the development of gated recursive unit (GRU) and long and short-term memory (LSTM) technology. However, for some long videos, GRU and LSTM cannot effectively capture long-term dependencies. This paper proposes a deep summarization network with auxiliary summarization losses to address this problem. We introduce an unsupervised auxiliary summarization loss module with LSTM and a swish activation function to capture the long-term dependencies for video summarization, which can be easily integrated with various networks. The proposed model is an unsupervised framework for deep reinforcement learning that does not depend on any labels or user interactions. Additionally, we implement a reward function (R(S)) that jointly considers the consistency, diversity, and representativeness of generated summaries. Furthermore, the proposed model is lightweight and can be successfully deployed on mobile devices and enhance the experience of mobile users and reduce pressure on server operations. We conducted experiments on two benchmark datasets and the results demonstrate that our proposed unsupervised approach can obtain better summaries than existing video summarization methods. Furthermore, the proposed algorithm can generate higher F scores with a nearly 6.3% increase on the SumMe dataset and a 2.2% increase on the TVSum dataset compared to the DR-DSN model.

## 1. Introduction

According to YouTube, there were approximately 5.5 billion daily video views in the first quarter of 2022. The massive amount of video information available causes people to spend significant time browsing and understanding redundant videos. Therefore, it is important to determine how to find relevant videos quickly among the endless video supply. Video retrieval techniques can help people find videos related to keywords, whereas video summarization techniques can extract representative information directly from a video. Video summarization techniques can generate concise summaries for videos that convey the important components of a complete video. Generating video summaries typically involves using a limited number of images or video clips to represent the main content of an original video sequence, which preserves the authenticity of the video information and saves a significant amount of space.

Research on video summarization techniques has been conducted for decades and good results have been achieved by traditional methods based on sparse modeling representative selection (SMRS) [1], DC programming [2], etc. With the development of deep learning, many researchers have used deep learning methods to extract features from videos and have achieved good results. M. Fei et al. [3] further introduced image entropy in deep learning to maintain the diversity of summaries. T. Hussain et al. [4] introduced aesthetic and entropic features to keep the summaries interesting and diverse. T. Hussain et al. [5] applied deep video summarization technology to real life. They proposed a video summarization method using depth features of the lens segmentation method and applied it to resource-constrained devices. In addition, T. Hussain et al. [6] proposed a deep learning-based video summarization strategy for industrial surveillance scenes to achieve coarse and refined video data, which provides a great contribution to the application of video summarization technology. Additional layers are meaningful in these methods [4,5,6], but they will consume more computation time and increase the complexity of the system. Therefore, an efficient learning mechanism to train DL models remains a challenge. Furthermore, it is difficult to handle the long-term dependency relationship due to the long timeline when processing the long-term monitoring data in the existing methods.

However, deep-learning-based methods rely on labels, whereas reinforcement learning does not rely on labels and allows models to explore and select features in an unsupervised manner. Reinforcement learning updates model parameters using reward functions and gradient descent techniques. In recent years, various researchers have combined deep learning with reinforcement learning to apply deep reinforcement learning (DRL) methods to the task of video summarization. Zhou et al. [7] proposed a deep learning framework that combines a bidirectional long short-term memory (LSTM) network (BiLSTM) with DRL and called their framework the diversity-representativeness deep summarization network (DR-DSN). The deep summarization network (DSN) predicts a probability for each video frame that indicates the likelihood of selecting that video frame and then takes action to select frames based on the predicted probability distribution to perform video summarization. DSN achieves good results and is competitive in the field of unsupervised video summarization. In addition, most deep-learning-based methods generate video summaries based on supervised learning, where they learn the importance of frames by modeling the temporal dependency between frames or the spatiotemporal structure of the video. The cost of producing video summarization datasets with labels is very expensive, so we focus on an unsupervised video summarization model based on deep reinforcement learning.

However, the performance of a deep reinforcement learning model degrades as video length increases because of the long-term dependency problem. In video summarization tasks, connections between adjacent frames are often established using recursive neural networks, but for tens of thousands of frames of video information, it is difficult to retain information from distant frames, which may lead to the degradation of model performance. This problem is called the long-term dependency problem. This problem limits the performance of existing approaches that focus on semantic objects, actions, emotions, and diversity. Even BiLSTM-based methods are not immune to the problems caused by long-term dependency. Large kernel sizes and deep networks give convolutional neural networks (CNNs) the ability to alleviate the long-term dependency problem. However, RNNs have a completely different structure compared to CNNs and we cannot directly employ the CNN structure in an RNN. To solve this problem, Trinh et al. [8] introduced auxiliary loss into the main supervised loss function to reconstruct or predict sequence information, which allows RNNs to capture long-term dependencies.

In this paper, to address the long-term dependency problem in the video summarization task, we equate video summarization to a sequential decision process based on inspiration from Trinh et al. [8]. We propose a summarization selection network with unsupervised summarization loss based on an infinite norm. Similar to existing DSNs [7], this network also utilizes an encoder–decoder architecture, where the encoder is a CNN that extracts high-dimensional features from videos and the decoder is a one-way LSTM network utilizing unsupervised summary loss to capture the long-term dependencies between video frames. The value of video frames can be calculated from video frame features. The higher the value, the greater the probability of being selected as a key frame. We trained the proposed network using an end-to-end reinforcement learning framework to solve the video summarization problem by updating the model parameters using a reward function that does not rely on any labels or user interactions. Finally, we propose a discrete degree reward function R(S) and compare it to other reward functions to demonstrate its practical value. An overview of the proposed model is presented in Figure 1. Compared to a traditional RNN, the proposed network is much better at maintaining long-term memory.

An unsupervised auxiliary summarization loss module is proposed to reconstruct random segments of a video by connecting to the LSTM network in parallel at randomly determined anchor points during model training. This loss is used to adjust the parameters of the model by calculating the degree of video feature loss during training. Experimental results demonstrate that the proposed unsupervised auxiliary summary loss can accurately capture the long-term dependencies in videos, significantly improving the performance and generalization ability of the LSTM network.

Furthermore, we explored several reward functions in reinforcement learning for the video summary model. The diversity-representation reward was proposed by Zhou et al. [7]. The diversity reward evaluates the degree of diversity of generated summaries by computing the differences between selected frames. The representation reward is used to measure the degree to which generated summaries can represent an original video. Although this reward can identify diversity-representative video frames, the cluttered distribution of excessive non-key frames disrupts summarized videos. To address this issue, we propose dispersion rewards, which allow the proposed method to cluster key frames together and reduce the probability of selecting non-key frames. As a result, our model preserves segment integrity and generates high-quality summaries. Additionally, our proposed method is generic and can be easily integrated into existing unsupervised video summarization models.

In summary, the main contributions of this paper are threefold.

We propose a DRL framework with an unsupervised summary loss for video summarization called AuDSN, which solves the long-term dependency problem of reinforcement-learning-based key frame extraction algorithms. Compared to RNN/ LSTM-based video summarization methods, our proposed video summarization framework can capture the long-term dependencies between video frames more effectively;We employ an unsupervised auxiliary summarization loss for video summarization, which assists in tuning network parameters by calculating the percentage differences between original and selected features. Additionally, the unsupervised auxiliary summarization loss does not increase the parameters of the model and can be easily integrated into other video summarization models;We propose a novel reward function called dispersion reward ($R_{dis}$) and employ it as a final reward function. Experimental results demonstrate the effectiveness of this reward function for video summarization.

The remainder of this paper is organized as follows. The related work is discussed in detail in Section 2. Section 3 describes the proposed pipeline-based AuDSN model. We present the experimental results in Section 4. Finally, Section 5 summarizes the key contributions of our work and concludes this paper.

## 2. Related Work

### 2.1. Video Summarization

Traditional video summarization methods [9,10,11] are shot- or segment-based methods, meaning an input video is divided into short shots or segments using various detection or segmentation algorithms. However, with the development of deep learning, the research on video summarization has made significant progress. Deep-learning-based video summarization utilizes neural networks to extract video information and select key segments and key frames in an end-to-end manner. Deep learning research on video summarization can be divided into three main categories: supervised video summarization [12,13,14,15,16], unsupervised video summarization [7,17,18,19,20], and weakly supervised video summarization. Our approach focuses on unsupervised video summarization.

Early supervised-learning-based approaches treated the selection of key frames as a prediction problem by modeling the temporal dependencies of each frame to estimate the importance of each frame and generate video summaries based on importance values. Zhang et al. [12] used LSTM to model the dependencies between video frames and derive representative and compact video summaries. One year later, Zhao et al. [13] proposed a hierarchical RNN based on LSTM that is more suitable for video summarization. Li et al. [15] proposed an efficient CNN based on global diverse attention by improving the self-attention mechanism of the transformer network, which adapts an attention mechanism from a global perspective to consider the pairwise temporal relationships between video frames.

In the context of unsupervised video summarization, because there is no universally accepted definition of video key frames, the goal of most existing unsupervised approaches is to maximize a viewer’s ability to grasp the original content of a video based on selected representative key frames. Mahasseni et al. [21] first applied a generative adversarial network (GAN) to the task of video summarization and their key concept was to train a deep summarization network that uses an unsupervised approach to minimize the distance between an input video and summarization distribution.

He et al. [17] proposed a conditional GAN based on self-attentiveness. The generator generates weighted frame features and predicts the importance of each frame, while the discriminator is used to distinguish between weighted frame features and original frame features. Rochan et al. [18] proposed a video summarization model based on a GAN and fully convolutional sequence network to perform video summarization utilizing unpaired data. However, due to the instability of the training stage and the limitations of the evaluation metrics of GAN methods, the GAN-based methods do not achieve good results.

Zhou et al. [7] used a combination of reinforcement learning and reward functions for video summarization. Video summarization was formulated as a sequential decision process, and a diversity-representativeness reward was used to train the summarizer to produce diverse and representative video summaries. In this model, BiLSTM is used as a decoder to estimate the importance of frames, but the LSTM forgetting mechanism leads to the exponential decay of information, limiting the model’s ability to capture long-term scale information. To address these issues, we introduce an unsupervised auxiliary summarization loss module to help train the LSTM network to capture the long-term dependencies between video frames. We also propose a novel reward with a dispersion reward. The dispersion reward function uses more intuitive criteria to help models improve performance.

### 2.2. DRL

It is common knowledge that deep learning has strong perceptual ability but lacks decision-making ability, whereas reinforcement learning has decision-making ability but lacks perceptual capabilities. Combining these two approaches can provide a solution to the perceptual decision problem in complex systems [22]. Therefore, DRL was devised to combine the perceptual ability of deep learning with the decision-making ability of reinforcement learning to achieve control directly based on input signals, which is an artificial intelligence method similar to the human thinking process [23]. In March of 2016, AlphaGo [24] defeated top professional player Lee Sedol with a score of 4:1 in a historic Go match after practicing and reinforcing itself over tens of thousands of games. This achievement was a testament to the potential of DRL development. Over the past few years, DRL has solved many real-world challenging decision problems with spectacular success. DRL has been successfully applied in the fields of computer vision, natural language processing, gaming, and robotics [25].

In the field of computer vision, DRL has been successfully applied to tasks such as landmark detection [26,27], object detection [28], object tracking [29], image registration [30], image segmentation [31], and video analysis [32]. Video summarization is a useful, but difficult task in video analysis that involves the prediction of objects or tasks in a video. The video summarization task can also be considered a decision-making task for each frame in a video. This task has been continuously improved since an unsupervised video summarization DRL method was first proposed by Zhou et al. [7]. Next, Zhou et al. [33] implemented a summarization network using deep Q-learning (DQSN) and used a trained classification network to provide rewards for training DQSN. This method is a weakly supervised approach based on reinforcement learning that utilizes easily accessible video-level category labels and encourages summaries to contain category-relevant information and maintain category identifiability. Liu et al. [34] used a 3D spatiotemporal U-net to encode the spatiotemporal information of input videos efficiently, allowing an RL again to learn from spatiotemporal information and predict the acceptance or rejection of video frames in a video summary. Unlike supervised learning methods, DRL learns information from the environment and does not require a large number of labeled data, which enables many machine learning applications for which no large labeled training data are available.

### 2.3. Long-Term Dependencies

In the field of artificial intelligence, many important applications require learning long-term dependencies between sequences [35]. It is challenging to provide neural networks with the ability to model the long-term dependencies in sequence data. In a video analysis, a 2-hour movie typically contains approximately 170,000 images. If we use deep learning to understand the content of a movie, long-term dependencies for 170,000 images must be constructed to capture the features of the movie. Typically, this is achieved using gradient descent and back-propagation through time (BPTT) in recursive networks. However, learning long-term dependencies using gradient descent is difficult because the gradients computed by BPTT tend to disappear or explode during training (Hochreiter et al. [36]). The solution to this problem is to mitigate vanishing gradients by allowing information to be stored in memory (LSTM [37]) or by using a short-circuiting mechanism in a residual structure (ResNet [38]). Additionally, for BPTT to work properly, intermediate hidden states in a sequence must be stored. However, the memory requirement is proportional to the sequence length, making it difficult to apply BPTT to long video sequences.

Vorontsov et al. [39] investigated the effects of the widening of the spectral boundary on convergence and performance by controlling the orthogonality constraint and regularization of the power matrix to solve the gradient disappearance and explosion problem associated with BPTT during the training process. This approach can capture long-term dependencies to some extent and improve the convergence speed of gradient descent. Trinh et al. [8] proposed an auxiliary loss for capturing long-term dependencies, which can help RNNs reconstruct the previous event in a sequence or predict the next event in a sequence. Based on the development of transformers, Dai et al. [40] introduced the concept of recursion in a transformer by reusing historical representations to propose the transformer-xl model. This model does not compute hidden states for each new segment but reuses hidden states obtained from previous segments. Reused hidden states serve as memory for the current segment and circular connections are established between segments. However, the transformer models have a large number of parameters and consume a large amount of computational power for training. In contrast to these existing approaches, our proposed video summarization method reconstructs the current video information between segments by incorporating unsupervised auxiliary summarization loss functions, which helps the sequence model capture long-term dependencies. It is noteworthy that the unsupervised auxiliary summarization loss function does not require the use of any labeling information and the auxiliary module only works in the training model and does not increase the overall parameters and complexity of the model.

## 3. Methodology

This paper proposes a deep summarization network with auxiliary summarization losses to improve the performance of video summarization. Specifically, the auxiliary summarization losses are used as auxiliary training in the proposed AuDSN model. A brief overview of the AuDSN model is detailed as follows.

We consider video summarization as the task of making decisions for each frame in a video. We implement DRL models to predict the value of the information contained in each frame and determine the probability of each frame being a key frame based on these values. We then select the final key frames based on a probability distribution. Inspired by Zhou et al. [7], we developed an end-to-end reinforcement-learning-based summarization network. The pipeline of this network is presented in Figure 2. We train AuDSN using reinforcement learning. AuDSN receives observations (video frames) at each time step and performs various actions on the observations. Additionally, AuDSN generates auxiliary summary loss values. The environment receives the actions and auxiliary summary loss values and generates a reward and observation for the next time step. The data ($O_{i},A_{i},O_{i}\_,R_{i}$) generated during this process are recorded in a sample database and when the sample data are sufficient, loss values are calculated and the model parameters are updated.

The proposed AuDSN is an encoder–decoder architecture, where the encoder uses a CNN (typically GoogLeNet [41]) to extract visual features ${\left\{x_{t}\right\}}_{t=1}^{T}$ from an input video frame sequence of length t ${\left\{v_{t}\right\}}_{t=1}^{T}$. The decoder uses an LSTM network that can capture temporal features. It randomly inserts anchor points in the LSTM network and introduces unsupervised auxiliary summarization loss. A fully connected layer serves as the final layer of the network. The LSTM network takes the high-dimensional features ${\left\{x_{t}\right\}}_{t=1}^{T}$ generated by the encoder as inputs and generates the corresponding hidden states ${\left\{h_{t}\right\}}_{t=1}^{T}$, some of which will be passed to the unsupervised auxiliary summarization loss function to help the summary network reconstruct historical information as a method of establishing long-term dependencies. The hidden state ${\left\{h_{t}\right\}}_{t=1}^{T}$ is loaded with past information. The value of each frame is obtained through a fully connected layer using the swish activation function [42] and the fully connected layer using the sigmoid activation function. The value of each frame is used to determine the action *a* as follows:(1)$$\;\mathrm{value}{\;}_{t}=R\left(w_{r}\times \sigma \left(w_{\sigma }\times h_{t}+b_{\sigma }\right)+b_{r}\right),$$
(2)$$p_{t}=\frac{\;\mathrm{value}{\;}_{t}}{\sum_{t=1}^{T}\;\mathrm{value}{\;}_{t}},$$
(3)$$a_{t}=\;\mathrm{Bernoulli}\;\left(p_{t}\right),$$
where R(·) is the rectified linear unit (ReLU) activation function, σ is the sigmoid activation function, $w_{r}$, $w_{x}$ are the weights of the fully connected layer, and $b_{r}$, $b_{x}$ are the biases of the fully connected layer. $p_{t}$ denotes the probability distributions over all possible action sequences. $a_{t}$ are binary samples drawn from the Bernoulli distribution indicating whether the i−th frame is selected. The Bernoulli distribution is parameterized by $p_{t}$. We obtain a video summary composed of the selected key frames $S=\{\cdots y_{a_{t}}\cdots \}$, where $y_{a_{t}}$ is the selected key frame with $a_{t}=1$. We only update the decoder during the training process.

### 3.1. Auxiliary Summarization Loss

To enhance the ability of LSTM to capture long-term dependencies, we propose a form of unsupervised auxiliary summarization loss that applies to video summarization. This loss not only enhances the memory capabilities of LSTM but can also be easily migrated to other models. It is experimentally demonstrated that this auxiliary summarization loss can help the summarization network establish longer-lasting dependencies and improve the performance of video summarization. During the training progress, we randomly sample multiple anchor locations in the LSTM and insert an unsupervised auxiliary summarization loss function at each location.

An unsupervised auxiliary summarization loss function first reconstructs memories from the past by sampling the subsequence after its anchor point, copying the hidden state of a subsequence of length n after its anchor point, and inserting the first input of this subsequence into the decoder network. The rest of the subsequence features are reconstructed using the decoder network, as shown in Figure 3. By using this training method, anchor points can be used as temporary memory points for the decoder network to store memories in a sequence. When we insert enough anchor points, the decoder network remembers the entire sequence and reconstructs memories. Therefore, when we reach the end of the sequence, the decoder network remembers a sufficient number of sequences and can generate a high-quality video summary.

The introduction of unsupervised auxiliary summarization loss in the decoder network results in the need for some additional hyperparameters, namely the sampling interval T and subsequence length N. The sampling interval is defined as the samples per unit length that are extracted from the decoder network and composed into subsequences. The subsequence length represents the number of original features contained in each subsequence. We define the auxiliary summarization loss as follows:(4)$$L_{\mathrm{auxiliary}\;}=\frac{\sum_{i=1}^{T}L_{i}}{\sum_{i=1}^{T}l_{i}},$$
where T denotes the sampling interval, $L_{i}$ denotes the loss evaluated on the *i*-th sampling segment, and the overall auxiliary summarization loss is calculated by summing all subsequence auxiliary summarization losses for a segment. We define the subsequence auxiliary summarization loss as follows:(5)$$L_{i}=\frac{1}{N\times l}\sum_{j=1}^{l}\sum_{i=1}^{N}\left|Y-C\right|,$$
where N denotes the subsequence length, *l* denotes the feature dimension (based on the CNN encoder, the feature dimension we obtain is typically 1024), *Y* denotes the original features of the subsequence, and *C* denotes the key summary features of the subsequence after decoder selection. The auxiliary summarization loss of each subsequence is obtained by computing the infinite norm error between the vectorized *Y* and *C* features. This auxiliary summarization loss represents the percentage of the variance between the original features *Y* and key summary features *C* of the subsequence selected by the decoder.

Tuning hyperparameters is a very expensive process, so we set the lengths of all selected subsequences N to be equal. Because the addition of unsupervised auxiliary summarization loss does not change the structure of the decoder module, and high-quality embeddings of input sequences can be learned based on the unsupervised auxiliary summarization loss, the weights of the LSTM network can be fine-tuned using the back-propagation step only.

### 3.2. Reward Function

In reinforcement learning, the goal of an agent is represented as a special signal called a reward, which is transmitted to the agent through the environment. At each moment, the reward is a single scalar value. Informally, the goal of an agent is to maximize the total reward it receives. This means that what needs to be maximized is not the current reward, but the long-term cumulative reward.

An agent always learns how to maximize its gain and the amount of gain is determined by the reward function. If we want an agent to perform a particular task, then we must provide rewards in such a manner that the agent maximizes its gain while completing the task. Therefore, it is critical that we design the reward function in a way that truly achieves our goal. In video summarization, the key frames are selected from the video, and the selected frames should be diverse, representative, and uniform. Gygli et al. [10] mentioned that there is no standard answer for a correct summary, but we must guarantee the generation of a high-quality summary. Figure 4 presents the results of the label visualization of three sample videos (Cooking, Fire Domino, and Jumps) from the SumMe dataset. Each video was labeled by 15 to 18 people and each line in the video visualization represents one person’s selection result. As shown in Figure 4, we should select key frames from labels such as those in Figure 4a,b for video summarization. In our proposed model, we demonstrate that combining dispersion rewards with two rewards for diversity and representativeness can produce enhanced summaries.

#### 3.2.1. Diversity Reward

The diversity reward function calculates the degree of diversity of a video summary by evaluating the dissimilarity between selected key frames. Specifically, it calculates the variability between selected video frames in the computational feature space to evaluate the diversity of a video summary quantitatively. The diversity reward function is defined as follows:(6)Rdiv=1yy−1∑t∈y∑t′∈yt≠t′d(xt,xt′),
where $y=\{y_{a_{t}}|a_{t}=1,t=1\cdots T\}$ is the set of selected video frames, $x_{t}$ is the feature sequence corresponding to the video frames and d(·,·) represents the dissimilarity function to calculate the dissimilarity between two frames, which is defined as follows:(7)$$d\left(x_{t},x_{t^{\prime }}\right)=1-\frac{x_{t}^{T}x_{t^{\prime }}}{{∥x_{t}∥}_{2}{∥x_{t^{\prime }}∥}_{2}}.$$

Therefore, the reward obtained by the agent is higher when the selected frames are more diverse (i.e., larger differences between selected video frames are favored). To ignore the similarity between two temporally distant frames, we set $d(x_{t},x_{t})=1$ if $\left|t\;-\;t’\right|\;>\;\lambda$, where λ controls the degree of temporal distance consideration [7].

#### 3.2.2. Representativeness Reward

The representativeness reward is used to measure how well selected key frames represent an original video [7]. Evaluating the representativeness of a video summary can be formulated as the k-medoids problem [43], where the representativeness reward is defined as follows:(8)$$R_{rep}=\exp\left(-\frac{1}{T}\sum_{t=1}^{T}\min_{t^{\prime }\in y}{∥x_{t}-x_{t^{\prime }}∥}_{2}\right),$$
where $x_{t}$ is the feature of the t−th frame and *y* is the set of selected frames, as described above.

#### 3.2.3. Dispersion Reward

When using representativeness rewards, the degree of representativeness is defined as a k-medoids problem. To ensure that the selected video frames are uniform, we propose a dispersion reward function to complement the representativeness reward function to achieve better selection results. The dispersion reward function uses more intuitive criteria. As shown in Figure 5, the distribution of (a), which has a higher dispersion reward, exhibits significantly better clustering than that of (b), which has a lower dispersion reward. We want clustering results to have small intra-class distances and large inter-class distances (i.e., classification results should have high discrimination). To this end, a criteria function reflecting both intra- and inter-class distances can be constructed.

The intra-class departure matrix is generated by calculating the distance from each sample point to the cluster center. The intra-class distance criteria function, which produces an intra-class departure matrix $S_{W}$, is defined as follows:(9)$$S_{W}^{j}=\frac{1}{n_{j}}\sum_{i=1}^{n_{j}^{s}}\left({\bar{x}}_{i}^{\left(j\right)}-{\bar{m}}_{j}\right){\left({\bar{x}}_{i}^{\left(j\right)}-{\bar{m}}_{j}\right)}^{T},j=1,2\ldots c,$$
where $\bar{m}$ is the sample mean vector of class *w*, sn denotes the number of samples in class *w*, and *c* denotes the number of categories classified. ${\bar{x}}_{i}^{\left(j\right)}$ denotes the $i^{th}$ data sample in the $j^{th}$ class.

The inter-class distance criterion function, which produces an inter-class deviation matrix $s_{b}$, is defined as follows:(10)$$S_{B}=\sum_{j=1}^{c}\frac{n_{j}^{s}}{N^{S}}\left({\bar{m}}_{j}-\bar{m}\right){\left({\bar{m}}_{j}-\bar{m}\right)}^{T},$$
where *c* denotes the number of classes to be classified, *m* is the mean vector of all samples to be classified, $n^{s}$ denotes the number of samples in class *w*, $N^{S}$ denotes the total number of samples, and $m_{j}$ denotes the class center of each class.

The dispersion reward based on the intra-class distance criteria function and inter-class distance criteria function is defined as follows:(11)$$R_{dis}=\mathrm{Tr}\left[S_{w}^{-1}S_{b}\right],$$
where Tr[⋯] is the matrix trace.

$R_{div}$, $R_{rep}$, and $R_{dis}$ complement each other and work jointly to guide the learning of AuDSN. The determination of optimal weights for the different reward types is discussed later in this paper. The basic reward function is expressed as follows:(12)$$R=R_{div}+R_{rep}+R_{dis}.$$

### 3.3. Deep Summarization Network with Auxiliary–Summarization Loss-Based Video Summarization

We train the summary network using a DRL framework. The goal of the proposed summarization method [7] is to learn a policy function $\pi_{\theta }$ with parameters θ by maximizing the expected rewards
(13)$$J\left(\theta \right)=E_{p_{\theta }\left(a_{1:T}\right)}\left[R\right],$$
where $p_{\theta }$ is computed using Equation (Equation 2) and *R* is computed using Equation (Equation 12). E represents the expected value for calculating the reward. $\pi_{\theta }$ is defined by our AuDSN.

The agent selects key frames based on the policy function $\pi_{\theta }$. This selection is defined as an action $a_{t}$. The agent combines the current observation state $o_{t}$ and feature space of the selected video frames to predict a new observation state $o_{i}\_$. The next action $a_{t+1}$ is determined based on the *n*-th reward $R_{n}$. The derivative of J(θ) is computed using the REINFORCE algorithm proposed by [44]. Only the encoder of the network is updated during training.

To reduce the computational load, the final gradient is calculated as the average gradient after executing *N* iterations on the same video. A constant baseline *b* is subtracted from the $n^{th}$ reward $R_{n}$ to avoid the type of high variance that leads to divergence. This baseline *b* is calculated based on the moving average of the rewards from the previous time steps. The gradient formula is defined as follows:(14)$$\nabla_{\theta }J\left(\theta \right)\approx \frac{1}{N}\sum_{n=1}^{N}\sum_{t=1}^{T}\left(R_{n}-b\right)\nabla_{\theta }\log\pi_{\theta }\left(a_{t}|h_{t}\right).$$

Additionally, we introduce a regularization term into the probability distributions to constrain the percentage of frames selected for the summary and add the $L_{2}$ regularization term to the weight parameters θ to avoid overfitting [7]. The parameters of the optimal policy function are optimized using the stochastic gradient algorithm by combining all conditions. We use Adam [45] as an optimization algorithm.

## 4. Experiments

### 4.1. Datasets

We primarily used the SumMe [10] and TVSum [46] datasets to evaluate our approach. SumMe consists of 25 videos ranging from 1 to 6 min in length with a variety of video content covering holidays, events, and sports captured from first- and third-person perspectives. Each video was annotated by 15 to 18 humans and the summary lengths range from 5 to 15% of the initial video durations. TVSum consists of 50 videos ranging from 1 to 11 min in length and contains video contents from 10 categories of the TRECVid Med dataset [47]. The TVSum videos were annotated by 20 users in the form of shot- and frame-level importance scores (from 1 to 5).

### 4.2. Evaluation Settings and Metrics

For fair comparison to other methods, we used the common F-score measure to assess the similarity between the selected key frames and ground-truth labels [10]. The F score is defined as follows:(15)$$F_{i}=\frac{1}{N_{i}}\sum_{j=1}^{N_{i}}2\frac{P_{i,j}R_{i,j}}{P_{i,j}+R_{i,j}},$$
where $N_{i}$ is the number of available user-generated summaries for the i-th test video $P_{i,j}$ and $R_{i,j}$ is the accuracy and recall ratio for the *j*-th user summary. These values are calculated on a per-frame basis. Accuracy refers to the degree of agreement between measured values and the corresponding “true” values. The recall ratio represents the total number of positive queries among the query sample, which can also be interpreted as the number of correct predictions among the true positive samples.

For a test video, we used a trained AuDSN to predict the value of each frame as importance scores. We computed shot-level scores by averaging frame-level scores within the same shot. For video shots, the kernel temporal segmentation (KTS) [48] method split the video into a set of non-intersecting temporal segments. The algorithm input the matrix of frame-to-frame similarities and output a set of optimal “change points” that correspond to the boundaries of temporal segments. A summary was generated by selecting the video shots with the highest total scores, which were calculated from the frame-level scores of the shots. The maximized total score problem is essentially the 0/1 knapsack problem, for which a near-optimal solution can be obtained using dynamic programming [7].

### 4.3. Implementation Details

To compare AuDSN to existing methods, we used a GoogleNet [41] model pre-trained on ImageNet to extract features and a 1024-dimensional feature vector from the penultimate layer of GoogleNet (pool 5) as the features for each video frame. We used the standard five-fold cross-validation in the training stage, meaning 80% of the videos were used for training and 20% were used for testing. We utilized the auxiliary summarization loss in our decoder module. The auxiliary summarization loss and reward mechanism were used to tune the parameters of the model and the KTS implementation [48] was used for temporal segmentation. During our experiments, we tested different activation functions to compare their effects on the model and kept adjusting the hyperparameters to achieve optimal performance.

### 4.4. Quantitative Results

In Table 1, we present the definitions of various models with different active functions and rewards. AuDSN corresponds to a deep summary network using unsupervised auxiliary summarization loss with a reward function using a diversity reward ($R_{div}$) and representativeness reward ($R_{rep}$). In practice, the swish activation function outperforms the ReLU activation function on some CNN-based tasks, such as image classification and machine translation. Therefore, we adopted swish as an activation function in AuDSN-S. AuDSN-D corresponds to deep summary networks using a diversity reward ($R_{div}$), representativeness reward ($R_{rep}$), and dispersion reward ($R_{dis}$). AuDSN-SD corresponds to a combination of swish as an activation function and all three reward functions.

Table 2 reports the performances of different models on the SumMe and TVSum datasets. One can see that AuDSN-SD performs significantly better than AuDSN-S and AuDSN-D on both datasets, indicating that using swish as an activation function and using $R_{div}$ + $R_{rep}$ + $R_{dis}$ summation as a reward function can better teach the summary network to generate high-quality video summaries. The performance of AuDSN-S indicates that using swish as an activation function improves performance by 0.4% (46.8%−46.4%) and 0.5% (59.3%−58.8%) on the two datasets, respectively, demonstrating that the swish activation function can provide the neural network layers with better performance for video summarization. The performance of AuDSN-D indicates that using $R_{d}is$ in combination with $R_{div}$ and $R_{rep}$ can improve performance on the SumMe and TVSum datasets by 0.2% (46.6%−46.4%) and 0.7% (59.5%−58.8%), respectively. These results demonstrate that adding the dispersion reward $R_{dis}$ to the reward function can improve model performance. The proposed method performs slightly better on the TVSum dataset than on the SumMe dataset. This may indicate that the dispersion reward function has a greater impact on longer videos.

In Table 3, we compare the proposed method to existing GRU- and LSTM-based methods using DRL. DR-DSN [7] is an unsupervised deep summarization method that uses BiLSTM as the decoder in its summarization network. DR-DSN${}_{\mathrm{SUP}}$ [7] is a supervised version of DR-DSN. DR1-DSN [49] uses Chebyshev distance instead of Euclidean distance in its diversity reward. DR2-DSN [49] is a version of DR1-DSN that uses a double hidden layer in its GRU. The results demonstrate that our proposed methods obtain higher scores than the previous methods on both datasets. These results indicate that our proposed unsupervised auxiliary summarization loss has stronger long-term dependency-capturing ability. The proposed AuDSN-SD yields higher F scores with a nearly 4.3% (47.7%−43.4%) increase on the SumMe dataset and 1.3% (59.8%−58.5%) increase on the TVSum dataset compared to the state-of-the-art DR2-DSN model [49]. Compared to DR-DSN [7], the proposed AuDSN-SD improves performance on the SumMe and TVSum datasets by 6.3% (47.7%−41.4%) and 2.2% (59.8%−57.6%), respectively. Additionally, we can see that our proposed method also outperforms the supervised model DR-DSN${}_{sup}$ (5.6% (47.7%−42.1%) improvement on SumMe and 1.7% (59.8%−58.1%) improvement on TVSum).

Table 4 reports the F-score results of the proposed method and state-of-the-art methods on both datasets. In this table, the performance of both supervised and unsupervised methods is reported. We can see that our proposed framework outperforms most of the other methods. The use of auxiliary summarization loss instead of bidirectional LSTM to capture long-term dependencies [7] yields a significant improvement in performance on both datasets. The results show that our proposed method can achieve competitive results compared to the state-of-the-art methods. It is worth noting that our proposed method is based on unsupervised learning, and the structure is simpler and lighter.

It is noteworthy that most of the previous classical baseline models are tens or even hundreds of megabytes in size. These models require a large number of parameters, large memory, high computing power, and a long time for training. The size of our proposed model is only 2.7 MB, which is much smaller than most previous models. In Table 5, we list the model sizes of some open-source models. The addition of auxiliary summarization loss allows our model to capture the dependencies between video frames accurately without using a complex network structure. Furthermore, the addition of auxiliary summarization loss does not increase the size of our model because auxiliary summarization loss is only applied in the training stage and automatically discarded at the end of training.

An extremely small number of parameters allows AuDSN to save significant training time and computational cost, which is important given the exponential growth in the number of online videos in recent years. AuDSN can significantly reduce the computational pressure on video websites so that they can handle larger batches of videos simultaneously. It is worth noting that our lightweight model also makes it possible to deploy video summarization on edge devices and mobile devices. When AuDSN is successfully deployed on mobile devices, it can improve the experience of mobile users and reduce the pressure on servers. Deploying AuDSN to edge devices can enrich the information processing capabilities of such devices and may give rise to novel application scenarios.

### 4.5. Comparison of the Augmented (A) and Transfer (T) Settings

Table 6 reports the F-score results of the proposed method in the canonical (C) augmented (A), and transfer (T) settings. C setting means that the training and testing sets are from the same dataset. A setting means that OVP, YouTube, TVSum, and 80% of SumMe are used as the training sets and the remaining 20% of SumMe is used as the testing set. T setting means that OVP, YouTube, and TVSum are used as the training sets and SumMe is used as the testing set. Experimental results show that using the A setting can improve the performance of our proposed model by 0.2%.

### 4.6. Qualitative Comparisons

In this section, we present qualitative comparison results for an example video called “playing ball” from the SumMe dataset. The frame labels of the video are presented in Figure 6. Figure 7 presents the qualitative results of our proposed models for the video. The gray bars in Figure 7 represent the results of user selection, the height of the bars represents the number of people who selected each frame as a key frame, and the colored bars represent the frame segments selected by the proposed methods. Some selected frames are presented at the bottom of each bar.

In general, our proposed models produce high-quality video summaries. The basic AuDSN model typically selects the high-quality parts of the summaries. AuDSN-SD using swish as an activation function with the combination of $R_{div}$ + $R_{rep}$ + $R_{dis}$ as reward functions can select frames with even higher values. Furthermore, our proposed AuDSN-SD does not produce jumps between frames, so the selected video summary is more coherent, making it easier for viewers to understand the video content without making them feel uncomfortable based on larges jumps between frames.

### 4.7. Ablation Experiments on the Effects of Hyperparameters

Table 7 presents the performance of AuDSN-SD when using different values for the hyperparameter T. The optimal hyperparameters were determined experimentally. One can see that the performance is optimal for the SumMe dataset when the sampling interval is 40 and for the TVSum dataset when the sampling interval is 30. The greater the video length, the better long-term dependencies can be captured by appropriately reducing the sampling interval.

Figure 8 presents the performance of AuDSN-SD when using different values for the hyperparameter N. We considered 10 to 90% of the video length as the subsequence length N to determine the optimal N. On the TVSum dataset, the model achieves the optimal performance when the subsequence length is 70% of the video length. On the SumMe dataset, the model achieves the optimal performance when the subsequence length is 50% of the video length. Model performance decreases when the subsequence length continues to increase. Therefore, as the video length increases, the selected subsequence length should also be increased appropriately to ensure optimal model performance.

### 4.8. Effects of different CNN Encoders

Table 8 presents the performance of AuDSN-SD using different encoders. We have attempted the most advanced feature extractor proposed in recent years to compare with our CNN encoder. From Table 6, we can see that using GoogleNet as the CNN encoder achieves the best performance. Furthermore, MobileNet, which can obtain lightweight features, is also a competitive encoder. We can give priority to using MobileNet as the CNN encoder of the model in the case of insufficient computational power.

### 4.9. Effects of Reward Weights

As shown in Table 3, the performance of AuDSN is improved by 0.2% (46.6%−46.4%) and 0.7% (59.5%−58.8%) when using discrete rewards on the SumMe and TVSum datasets, respectively. The main contribution of the discrete reward is improved video summary quality. The discrete reward attempts to group interrelated segments together. Therefore, the model with the discrete reward preserves the story line of the video by eliminating redundant jumps between adjacent clips. The comparison in Figure 7 reveals that the model with the dispersion reward outputs more continuous clips and fewer single video frames.

To allow our model to achieve better results, we evaluated the proposed AuDSN model by assigning different weights to the three reward functions. For the TVSum and SumMe datasets, we present several results based on different weights in Table 9 and visualize these results in Figure 9. One can see that representativeness rewards have a greater impact on the overall reward function than diversity rewards. In Figure 9, we present the results of the optimal $R_{div}$ = 0.2, $R_{rep}$ = 0.2, and $R_{rep}$ = 0.6 weights for the SumMe dataset, and $R_{div}$ = 0.4, $R_{rep}$ = 0.2, and $R_{rep}$ = 0.6 weights for the TVSum dataset. The three weights should satisfy: $R_{div}+R_{rep}+R_{dis}=1$. Therefore, the edges of the figure are triangular.

In Table 10, one can see that the best results are obtained when the dispersion reward weight is 0.2 for the TVSum Dataset. For long videos, the representativeness and diversity rewards play a greater role in the selection of summary segments/frames. The dispersion reward plays an auxiliary role and its purpose is to make video summaries more uniform and continuous while avoiding jumps between video frames and maintaining the continuity and integrity of the story line of a summary. The dispersion reward function does not improve model performance significantly, but it has a noticeable impact on the quality of video summaries.

## 5. Conclusions

In this paper, we proposed AuDSN, which is a deep reinforcement network model with unsupervised auxiliary summarization loss. We introduced unsupervised auxiliary summarization loss in the decoder and explored a novel reward function with a dispersion reward. Experimental results on two datasets (SumMe and TVSum) demonstrated that introducing unsupervised auxiliary summarization loss can improve the long-term dependency-capturing ability for a deep summarization network. Additionally, the swish activation function and dispersion reward function can help a deep summarization network construct more coherent, diverse, and representative video summaries. Furthermore, AuDSN is a very lightweight model with a size of only 2.7 MB, presenting the opportunity of deploying it on low-computing-power edge devices. In future work, we will also try to incorporate multi-information, such as sound features, into the model and explore multi-information versions of the deep summarization network framework. Furthermore, we will try to incorporate the proposed video summarization approach into modern media tools to make practical use of the proposed summarization algorithms.

## Figures and Tables

**Figure 1 sensors-22-07689-f001:**
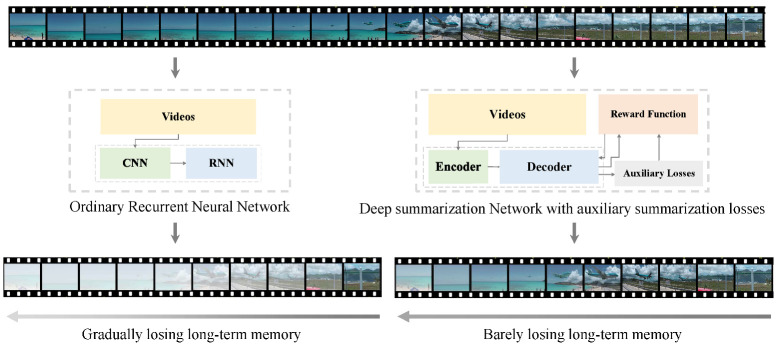
Overview of networks with long-term dependency capture capability (the proposed model) and recurrent neural networks.

**Figure 2 sensors-22-07689-f002:**
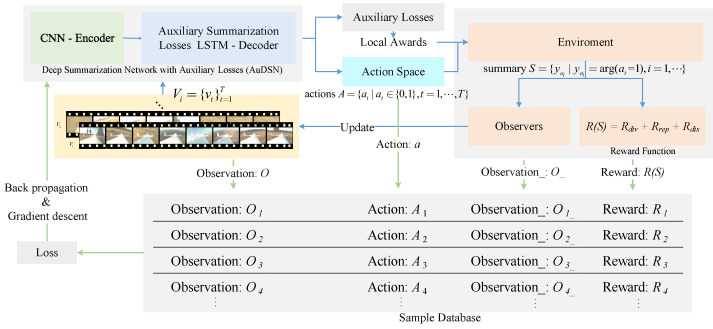
Pipeline of the proposed AuDSN. The blue arrow represents the process of generating a summary and the green arrow represents the process of model training.

**Figure 3 sensors-22-07689-f003:**
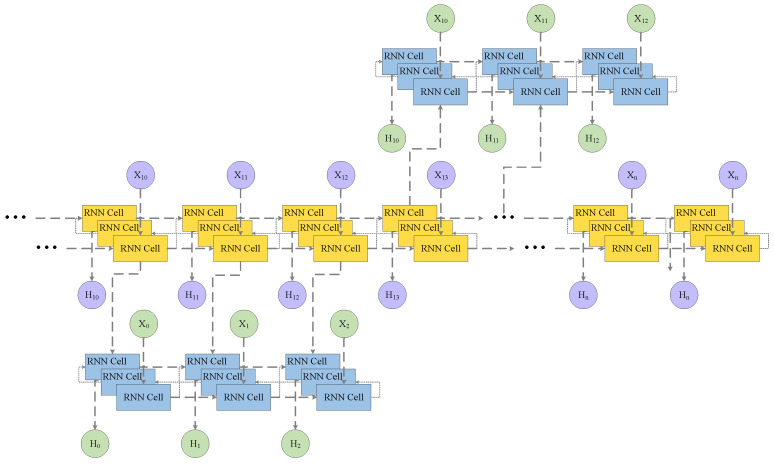
The decoder module with unsupervised auxiliary summarization loss. Yellow RNN cells represent decoder networks, blue RNN cells represent auxiliary loss modules. X represents the input features. H represents the decision of RNN cells based on input.

**Figure 4 sensors-22-07689-f004:**
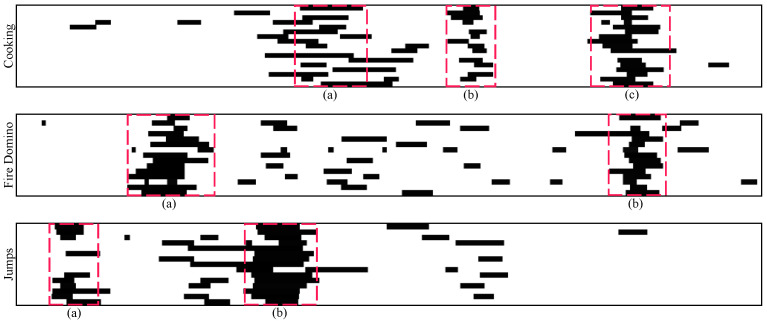
Visual results of human-selected labels for three example videos (cooking, fire domino, and jumps) from SumMe dataset. (**a**–**c**) are example labels which are selected by most people.

**Figure 5 sensors-22-07689-f005:**
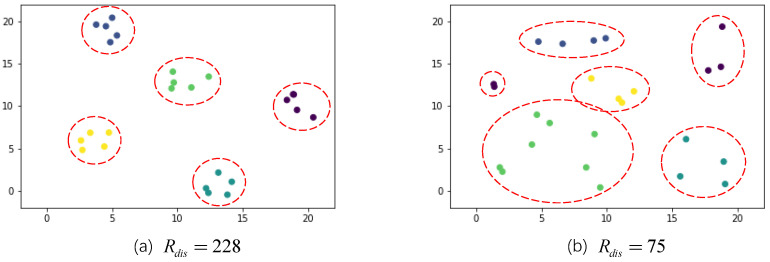
The dispersion reward results for two different clustering cases; (**a**) 20 artificially generated points; (**b**) 20 randomly generated points in a certain range. Different colors of points represent different classes.

**Figure 6 sensors-22-07689-f006:**
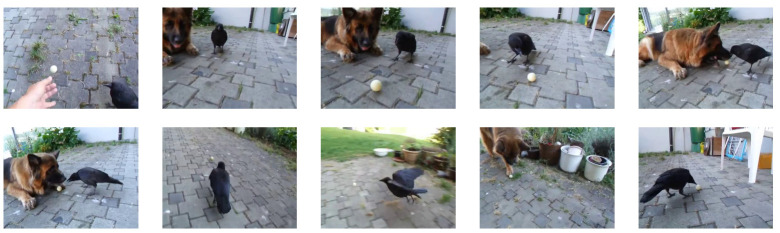
Examples of frame labels from the “playing ball” video.

**Figure 7 sensors-22-07689-f007:**
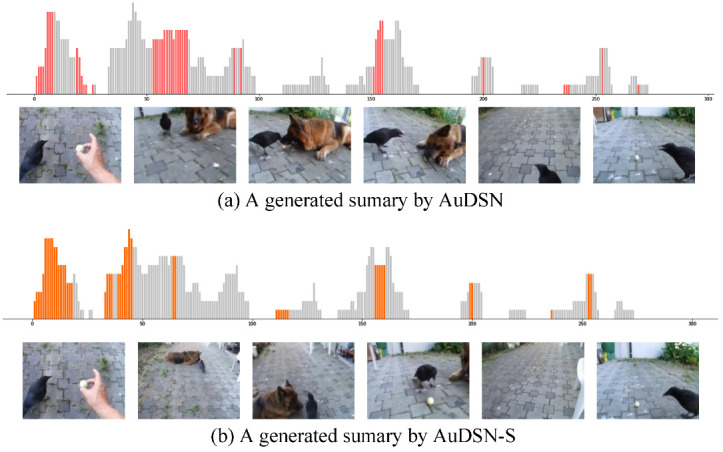
Video summaries generated by all variants of the proposed method.

**Figure 8 sensors-22-07689-f008:**
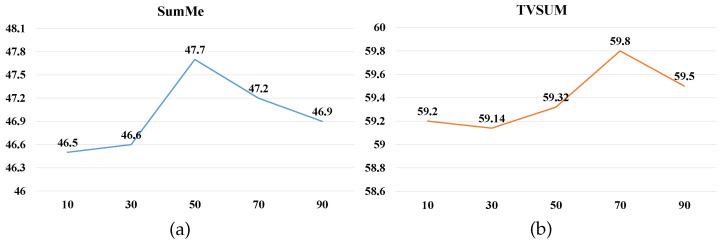
Performance with different subsequence lengths N on two datasets: (**a**) SumMe dataset and (**b**) TVSum dataset. The x-axis represents the percentage of video length and the y-axis represents the F score.

**Figure 9 sensors-22-07689-f009:**
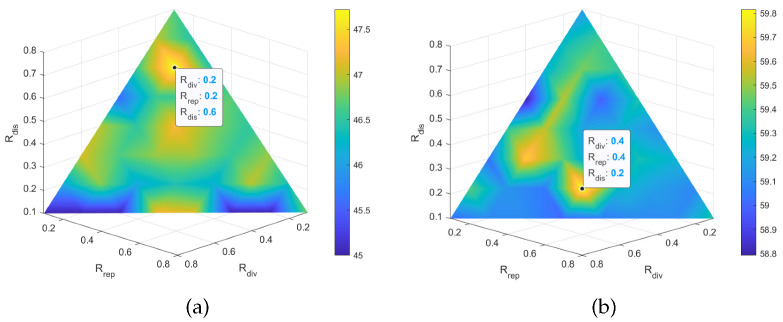
Effects of different reward weights on SumMe and TVSum performances (F score): (**a**) SumMe dataset and (**b**) TVSum dataset. The three axes correspond to the weights of the three reward functions and the depth of the colors represents the F scores.

**Table 1 sensors-22-07689-t001:** Definitions of different models.

Model Name	Activation Function	Reward
AuDSN	ReLU	$R_{div}$ + $R_{rep}$
AuDSN-S	swish	$R_{div}$ + $R_{rep}$
AuDSN-D	ReLU	$R_{div}$ + $R_{rep}$ + $R_{dis}$
AuDSN-SD	swish	$R_{div}$ + $R_{rep}$ + $R_{dis}$

**Table 2 sensors-22-07689-t002:** Performance (F-scores (%)) of different variants of the proposed method.

Method	SumMe	TVSum
AuDSN	46.4	58.8
AuDSN-S	46.8	59.3
AuDSN-D	46.6	59.5
AuDSN-SD	**47.7**	**59.8**

**Table 3 sensors-22-07689-t003:** Performance (F-scores (%)) of the proposed methods compared to other DRL-based methods.

Method	Network	SumMe	TVSum
DR-DSN [7]	LSTM	41.4	57.6
DR-DSN${}_{sup}$ [7]	LSTM	42.1	58.1
DR1-DSN [49]	GRU	42.9	58.3
DR2-DSN [49]	GRU	43.4	58.5
AuDSN [OURS]	LSTM	46.4	58.8
AuDSN-S [OURS]	LSTM	46.8	59.3
AuDSN-D [OURS]	LSTM	46.6	59.5
AuDSN-SD [OURS]	LSTM	**47.7**	**59.8**

**Table 4 sensors-22-07689-t004:** Performance (F-scores (%)) of the proposed method compared to existing methods.

Method	SumMe		TVSum	
	F1	Rank	F1	Rank
Random summary [50]	40.2	14	54.4	14
Online Motion-AE [51]	37.7	15	51.5	15
DR-DSN [7]	41.4	13	57.6	10
CSNet [52]	**51.3**	1	58.8	4
Cycle-SUM [53]	41.9	12	57.6	10
DR1-DSN [49]	42.9	9	58.3	8
DR2-DSN [49]	43.4	8	58.5	5
UnpairedVSN [18]	47.5	3	55.6	13
EDSN [54]	42.6	11	57.3	12
PCDL [55]	42.7	10	58.4	7
ACGAN [17]	46.0	5	58.5	5
DHAVS${}_{sup}$ [56]	45.6	6	60.8	2
3DST-Unet${}_{sup}$ [34]	47.4	4	58.3	8
M-AVS${}_{sup}$ [57]	44.4	7	**61.0**	1
AuDSN-SD [OURS]	47.7	2	59.8	3

**Table 5 sensors-22-07689-t005:** Sizes of the parameters generated by different methods.

Method	Model Size (MB)
DR-DSN [7]	10.0
CSNet [52]	28.0
UnpairedVSN [18]	95.4
AuDSN [OURS]	**2.7**

**Table 6 sensors-22-07689-t006:** F-score (%) of the LSTM-based approaches on SumMe and TVSum in the canonical (C), augmented (A), and transfer (T) settings.

Method	SumMe			TVSum		
	C	A	T	C	A	T
SUM-GAN${}_{rep}$ [21]	38.5	42.5	-	51.9	59.3	-
SUM-GAN${}_{dpp}$ [21]	39.1	43.4	-	51.7	59.5	-
DR-DSN [7]	41.4	42.8	42.4	57.6	58.4	57.8
CSNet [52]	**51.3**	**52.1**	45.1	58.8	59	**59.2**
Cycle-SUM [53]	41.9	-	-	57.6	-	-
AuDSN-SD [OURS]	47.7	47.9	**45.1**	**59.8**	**59.7**	57.8

**Table 7 sensors-22-07689-t007:** Performance with different sampling intervals T on two datasets.

T	10	20	30	40	50
SumMe	46.7	46.9	47.1	47.7	46.8
TVSum	59.3	59.3	59.8	59.3	58.7

**Table 8 sensors-22-07689-t008:** Performances with different CNN encoders on two datasets.

	GoogleNet	EfficientNet	MobileNet	ResNet	VGG	VIT
	[41]	[58]	[59]	[38]	[60]	[61]
SumMe	47.7	46.8	47.3	47.2	46.9	46.8
TVSum	59.8	59.3	59.5	59.4	59.4	59.2

**Table 9 sensors-22-07689-t009:** Effects of different reward weights on SumMe and TVSum performance (F-scores).

$R_{\mathrm{div}}$	$R_{\mathrm{rep}}$	$R_{\mathrm{dis}}$	SumMe	TVSum
0.1	0.1	0.8	46.7	59.1
0.1	0.2	0.7	46.4	59.4
0.1	0.3	0.6	46.4	59.4
0.1	0.4	0.5	46.7	59.3
0.1	0.5	0.4	46.5	59.1
0.1	0.6	0.3	46.9	59.2
0.1	0.7	0.2	46.4	59.1
0.1	0.8	0.1	46.8	59.3
0.2	0.1	0.7	47.0	59.3
0.2	0.2	0.6	**47.7**	59.5
0.2	0.3	0.5	46.6	59.0
0.2	0.4	0.4	46.8	59.2
0.2	0.5	0.3	46.5	59.3
0.2	0.6	0.2	47.0	59.2
0.3	0.1	0.6	46.6	59.2
0.3	0.2	0.5	46.6	59.5
0.3	0.3	0.4	47.2	59.2
0.3	0.4	0.3	46.9	59.0
0.3	0.5	0.2	46.2	59.3
0.4	0.1	0.5	45.6	58.8
0.4	0.2	0.4	46.5	59.6
0.4	0.3	0.3	46.9	59.2
0.4	0.4	0.2	46.4	**59.8**
0.4	0.5	0.1	47.1	59.1
0.5	0.1	0.4	47.0	59.4
0.5	0.2	0.3	46.6	59.2
0.5	0.3	0.2	46.1	59.0
0.5	0.4	0.1	47.2	59.0
0.6	0.1	0.3	47.1	59.0
0.6	0.2	0.2	46.8	59.1
0.7	0.1	0.2	46.1	59.4
0.8	0.1	0.1	46.5	59.0

**Table 10 sensors-22-07689-t010:** Weights that yield the best performances for the reward functions on both datasets.

DataSet	$R_{\mathrm{div}}$	$R_{\mathrm{rep}}$	$R_{\mathrm{dis}}$	F1
SumMe	0.20	0.20	0.60	47.7
TVSum	0.40	0.40	0.20	59.8

## Data Availability

Not applicable.

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
