# Peer review of "A Video Summarization Model Based on Deep Reinforcement Learning with Long-Term Dependency"

_sensors, 2022, doi:10.3390/s22197689_

Round 1

Reviewer 1 Report

Paper is interesting and well written. Some elements must be improved as indicated below:

Intro: CNN and RNN are indicated as methods to deal with sequences. Currently in many applications this combination is replaced by transformers. So you need to motivate why that is not a good choice here. In related work transformers do come back.

P4: for [13] no limitations are given. So why is a new method needed? Or does “in this model” refer to all the above methods?

The proposed method is based on reinforcement learning, yet it bears a lot of similarity to methods which use the same criteria like diversity that you have but than as loss functions. Explain in more detail what the differences are. Is it only the optimization function that is different? Are the criteria expressed in a different way?

P11: the KTS measure is unclear. Elaborate on its definition. When reading the rest of the paper I see KTS is only coming back in segmentation into shots but where is it used as an evaluation criterion?

4.3. Use of Googlenet already mentioned on p7 so refer to it rather than repeating

P12: the methods you compare to seem at least 3 years old. How does your method compare to more recent methods on these benchmark datasets?

Table 5 can be integrated with table 4 and in table 4 you can remove the ranking which doesn't add much.

Table 7: index column can be removed as it doesn't add anything as it is not combing back somewhere else.

Some typos:

P2: since -> because of

P6: to develop -> to improve

P7: continues reconstructing

Figure 2: updata -> update 

Reviewer 2 Report

A video summarization model based on deep reinforcement learning with long-term dependency

This paper proposes a deep reinforcement network model with unsupervised auxiliary summarization loss for video summarization. The authors proposed a reward function with a dispersion reward in unsupervised auxiliary summarization loss.

The idea of combining deep learning and reinforcement learning to benefit both perceptual ability and decision-making ability is perfect that used in this paper.

Also, the paper structure and its readability are suitable. (some minor needs)

Experimental is evaluated with two datasets SumMe and TVSum. There are appropriate evaluations, and both Quantitative and Qualitative comparisons show that the proposed idea makes sense.

Minors

Fig.3 needs more description and clarification.
Eq. 7 needs to introduce and describe its motivation to use.
In dispersion reward, eq. 9 needs more info. Where does it come from? Why this eq.?
The caption of Figure 6 is not clear and needs more description.

Round 2

Reviewer 1 Report

The answer on the reinforcement part vs deep learning is not answered well (but maybe also because the remark was not clear enough). What I was asking for was a conceptual comparison to classification based deep learning methods that use similar criteria. Focus of the current answer is to compare with other reinforcement based learning methods. Adding some more on this in the paper would be helfpul.
